# DRAFT, SKETCH, AND PROVE: GUIDING FORMAL THEOREM PROVERS WITH INFORMAL PROOFS

**Albert Q. Jiang**[1,2,†]       **Sean Welleck**[3,4,†]       **Jin Peng Zhou**[5,6,†]

**Wenda Li**[2]       **Jiacheng Liu**[3]       **Mateja Jamnik**[2]

**Timothée Lacroix**[1]       **Guillaume Lample**[1,‡]       **Yuhuai Wu**[5,7,‡]

[1]Meta AI   [2]University of Cambridge   [3]University of Washington   [4]Allen Institute for AI
[5]Google Research   [6]Cornell University   [7]Stanford University

## ABSTRACT

The formalization of existing mathematical proofs is a notoriously difficult process. Despite decades of research on automation and proof assistants, writing formal proofs remains arduous and only accessible to a few experts. While previous studies to automate formalization focused on powerful search algorithms, no attempts were made to take advantage of available informal proofs. In this work, we introduce *Draft, Sketch, and Prove (DSP)*, a method that maps informal proofs to formal proof sketches, and uses the sketches to guide an automated prover by directing its search to easier sub-problems. We investigate two relevant setups where informal proofs are either written by humans or generated by a language model. Our experiments and ablation studies show that large language models are able to produce well-structured formal sketches that follow the same reasoning steps as the informal proofs. Guiding an automated prover with these sketches enhances its performance from 20.9% to 39.3% on a collection of mathematical competition problems.

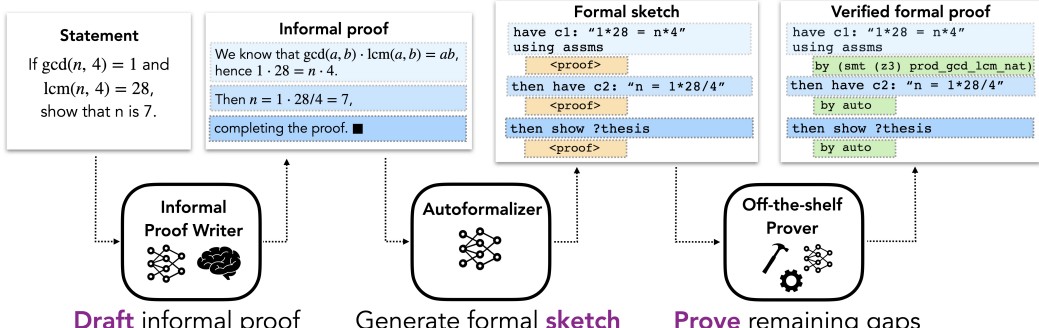

Figure 1: **Draft, Sketch, and Prove.** Starting with an informal statement, our framework yields a formal proof through a three-stage process: drafting informal proofs, mapping them into formal sketches, and proving the remaining conjectures. Concretely, an informal statement is a mathematical problem described in a mixture of natural and mathematical languages (e.g., formulae in LaTeX). Then, we use a large language model to autoformalize each informal proof into a formal sketch, which is a skeleton of the formal proof with open conjectures left unproven (indicated by the `<proof>` blocks). The formal sketch mirrors the structure of the informal proof. Finally, the open conjectures/gaps inside each formal sketch are proved by an off-the-shelf prover.

---

[†]Equal contributions as leading authors. Correspondence to: `qj213@cam.ac.uk`.
[‡]Equal contributions as senior authors.

## 1 INTRODUCTION

Formal proof automation is a challenging task that has been the focus of increased attention in recent years (Bansal et al., 2019b; Polu & Sutskever, 2020; Lample et al., 2022; Jiang et al., 2022; Wu et al., 2022). However, deep learning approaches have not been as successful as in other domains, mainly because of the scarcity of formal data. Indeed, formalizing proofs is notoriously difficult and only accessible to a handful of experts, which makes large annotation endeavors unrealistic (Wiedijk, 2008). The largest formal proof corpus is written in Isabelle (Paulson, 1994), and amounts to less than 0.6 GB in size, orders of magnitude smaller than datasets commonly used in vision (Deng et al., 2009) or natural language processing (Brown et al., 2020). To address the scarcity of formal proofs, previous studies have proposed to use synthetic data (Wu et al., 2021b), self-supervision (Polu & Sutskever, 2020; Han et al., 2022), or reinforcement learning (Bansal et al., 2019a; Polu et al., 2022) to synthesize additional formal training data. Although these methods alleviate the data insufficiency to some degree, none are able to capitalize on the bulk of human-written mathematical proofs.

Unlike formal mathematics, informal mathematical data is abundant and widely available. Recently, large language models trained on informal mathematical data showcased impressive quantitative reasoning abilities (Lewkowycz et al., 2022; Welleck et al., 2022). However, they often generate erroneous proofs and it is challenging to detect the faulty reasoning in these proofs automatically. Our work devises a novel approach called *Draft, Sketch, and Prove (DSP)* to translate informal mathematical proofs into formal ones and thus enjoy both the logical rigor provided by formal systems and the wealth of informal data. We give a schematic diagram of the *DSP* method in Figure 1 and describe it in Section 3. Recent work (Wu et al., 2022) demonstrates the feasibility of automatically translating informal statements into formal ones with large language models. *DSP* goes beyond and leverages large language models to generate *formal proof sketches* (Wiedijk, 2003) from *informal proofs*. Proof sketches consist of high-level reasoning steps that can be interpreted by formal systems such as interactive theorem provers. They differ from complete formal proofs in that they contain sequences of intermediate conjectures without justification. An example of informal proof with its corresponding formal proof sketch is provided in Figure 2. In the last step of *DSP*, we elaborate the formal proof sketch into a full formal proof using an automated prover to prove all intermediate conjectures.

We perform experiments to generate formal proofs of problems from the miniF2F dataset (Zheng et al., 2022) and show that a large portion of theorems can be proved automatically with this method. We investigate two settings where the informal proofs are either written by humans or drafted by a large language model trained on mathematical text. These two settings correspond to situations frequently occurring during the formalization of existing theories, where informal proofs are usually available, but sometimes left as exercises to the reader or missing due to space limits in the margin.

**Contributions:**

- We introduce a novel approach to leverage informal proofs to guide automated provers with formal proof sketches.
- To evaluate our approach, we build a dataset of manually curated informal statements and informal proofs aligned with formal statements in the miniF2F dataset (Zheng et al., 2022).
- We increase the proportion of problems solved by an automated prover on miniF2F from 20.9% to 38.9% given language-model-generated informal proofs, and up to 39.3% when proofs are written by humans.
- Through three ablation studies, we demonstrate the performance benefit of drafting informal proofs, annotating sketches with informal segments, and using automated provers to close open conjectures for the autoformalization of proofs.

## 2 BACKGROUND AND RELATED WORK

**Interactive theorem proving**    Modern verification systems for mathematics are centered around *interactive theorem provers (ITPs)*, such as Isabelle (Paulson, 1994), Lean (Moura et al., 2015), Coq (Barras et al., 1997), or Metamath (Megill & Wheeler, 2019). ITPs embed the mathematical definitions and theorems onto a solid logical foundation (e.g., Higher-Order Logic, Dependent Type Theory) implemented by their kernels. Every theorem must be checked by the kernel to be recognized by the ITP. To be proved formally, a theorem is first stated in the ITP's programming language, and

iteratively simplified into simpler objectives (or subgoals), until it can be reduced to already proven facts. In this paper, we will refer to proofs verified by a formal theorem prover as *formal proofs*, and proofs written in "standard" mathematics (e.g. in LaTeX) as *informal proofs*.

**Machine learning for formal proof synthesis**   Several approaches propose to combine machine learning with modern interactive theorem provers (Yang & Deng, 2019; Gauthier et al., 2021), and build upon the recent success of language models (Polu & Sutskever, 2020; Han et al., 2022; Polu et al., 2022; Jiang et al., 2022; Lample et al., 2022). These methods typically rely on sequence-to-sequence models (Sutskever et al., 2014) to generate the next step of a proof given the current proof state and perform search over the generated subgoals using powerful search methods such as MCTS (Silver et al., 2018; Wu et al., 2021a; Laurent & Platzer, 2022). Because search is computationally expensive, these language models are relatively small (with fewer than 1 billion parameters). Our method contrasts with these approaches in that we use a significantly reduced number of calls to the models, but also much larger language models (with up to 175 billion parameters) that showcase outstanding few-shot learning abilities (Brown et al., 2020).

**Machine learning for informal reasoning**   Language models have also been used in the context of purely informal mathematics (Lample & Charton, 2020; Hendrycks et al., 2021; Welleck et al., 2021; Drori et al., 2022; Welleck et al., 2022). Nevertheless, Lewkowycz et al. (2022) note that for quantitative question answering, models are prone to generate false positives: the model guesses the right answer while providing an incorrect proof. These errors are hard to spot without human inspection. Worryingly, the frequency of false positives increases with the difficulty of the problem. Our method builds on these findings and translates informal proofs into formal proofs. Since ITPs are logically grounded, once a formal proof is checked by them, we are guaranteed its correctness.

**Autoformalization**   In a position paper, Szegedy (2020) argued for attaining formal mathematical data from informal sources with neural networks. Wang et al. (2020) performed preliminary experiments where the evaluation was limited to text-level similarities on synthetic datasets. Recently, Wu et al. (2022) found that large language models (Chen et al., 2021; Chowdhery et al., 2022) are capable of few-shot *statement autoformalization*. Namely, a small number of examples are enough for them to learn to perform informal-to-formal translation of statements. In this paper, we investigate whether these findings can generalize to *proof autoformalization*, i.e., whether large language models can be used to translate informal proofs into formal ones.

## 3   METHOD

In this section, we describe our *Draft, Sketch, and Prove* (*DSP*) method for formal proof automation, which leverages informal proofs to guide automated formal theorem provers with proof sketches. We assume that each problem comes with an informal statement and a formal statement describing the problem. Our pipeline consists of three stages (depicted in Figure 1), which we present below.

### 3.1   DRAFTING INFORMAL PROOFS

The initial phase of the *DSP* method consists in finding informal proofs for a problem according to its description in natural mathematical language (possibly with LaTeX). The resulting informal proof is seen as a *draft* for the subsequent phases. In mathematical textbooks, proofs of theorems are in general provided, but are sometimes missing or incomplete. Therefore, we consider two settings corresponding to the presence or absence of the informal proofs. In the first, we assume that a "ground-truth" informal proof (i.e., one written by a human) is available, which is the typical scenario in the practice of formalizing existing mathematical theories. In the second setting, we make a more general assumption that the ground-truth informal proof is not given, and draft proof candidates with a large language model trained on informal mathematical data. The language model removes the dependence on human proofs and can produce multiple alternative solutions for every problem. Although there is no easy way to automatically verify the correctness of these proofs, the informal proof only needs to be *useful* for producing a sketch in the next stage.

### 3.2   MAPPING INFORMAL PROOFS INTO FORMAL SKETCHES

A formal proof sketch encodes the structure of a solution and leaves out low-level details (Wiedijk, 2003). Intuitively, it is a partial proof that outlines high-level conjecture statements. A concrete example of a proof sketch is shown in Figure 2. Although informal proofs often leave aside low-level details, (e.g., by stating their triviality), these details cannot be discharged in a formal proof, making

**Informal Statement:** Show that for any real number $a$, $10a \leq 28a^2 + 1$.

- - - - - - - - - - - - - - - - - - - - - - - - - - - - - - - - - - - - - - - - - - -

**Informal Proof:**
It suffices to show $0 \leq 28a^2 - 10a + 1$. First, consider completing the square for $28a^2 - 10a$ and observe that $\left(a - \frac{5}{28}\right)^2 = a^2 - \frac{10}{28}a + (5/28)^2$. Since $0 \leq \left(a - \frac{5}{28}\right)^2$, we get $0 \leq a^2 - \frac{10}{28}a + (5/28)^2$. Multiplying by 28 and simplifying gives $0 \leq 28a^2 - 10a + (25/28)$. Since $25/28 < 1$, the result follows.

- - - - - - - - - - - - - - - - - - - - - - - - - - - - - - - - - - - - - - - - - - -

**Formal Proof Sketch:**

```
theorem algebra_binomnegdiscrineq_10alt28asqp1:
  fixes a :: real
  shows "10 * a ≤ 28 * a^2 + 1"
proof - (* it suffices to show 0 <= 28a^2 - 10a + 1 *)
  have c0: "0 ≤ 28a^2 - 10a + 1"
  proof - (* observe that (a - (5/28))^2 = a^2 - (10/28)a + (5/28)^2 *)
    have c1: "(a - (5/28))^2 = a^2 - 10/28a + (5/28)^2" < ··· >
    (* we get 0 <= a^2 - (10/28)a + (5/28)^2 *)
    have c2: "0 ≤ a^2 - 10/28a + (5/28)^2" using c1 < ··· >
    (* Multiplying by 28 and simplifying gives 0 <= 28a^2 - 10a + (25/28) *)
    have c3: "0 ≤ 28a^2 - 10a + 28((5/28)^2)" using c2 < ··· >
    have c4: "0 ≤ 28a^2 - 10a + 28((5/28)*(5/28))" using c3 < ··· >
    have c5: "0 ≤ 28a^2 - 10a + (25/28)" using c4 < ··· >
    (* Since 25/28 < 1, the result follows. *)
    show ?thesis using c5 < ··· >
  qed
  show ?thesis < ··· >
qed
```

Figure 2: **A proof sketch in Isabelle.** The problem "Show that for any real number $a$, $10a \leq 28a^2 + 1$" is given with an informal proof and an associated formal proof sketch. The sketch first rewrites the original statement (`c0`), which is proved through 5 intermediary conjectures (`c1`..`c5`). We use a special token ($< \cdots >$) to indicate that the conjecture is "open" and should be tackled by an automated prover later. To facilitate the alignment between the informal and formal languages, we annotate the formal proof sketch examples with informal proof segments (shown in red), which are immediately followed by their formal counterparts.

straightforward informal-to-formal proof translation difficult. Instead, we propose to map informal proofs to formal proof sketches that share the same high-level structures. The low-level details missing from a proof sketch can later be filled by an automated prover. Since large informal-formal parallel corpora do not exist, standard machine translation methods are unsuitable for this task. Rather, we use the few-shot learning abilities of a large language model. Specifically, we prompt the model with a few example pairs containing *informal proofs* and their corresponding *formal sketches*, followed by an informal proof yet to be translated. We then let the model generate the subsequent tokens to obtain the desired formal sketch. We refer to this model as an autoformalizer.

### 3.3 PROVING OPEN CONJECTURES IN THE SKETCHES

As the last part of the process, we execute off-the-shelf automated provers to fill in the missing details in proof sketches, where "automated provers" refers to systems capable of producing formally verifiable proofs. Our framework is agnostic to the specific choice of the automated prover: it can be symbolic provers such as heuristic proof automation tools, neural-network-based provers, or hybrid approaches. If the automated prover successfully closes all the gaps in the proof sketch, it returns the final formal proof which can be checked against the problem's specification. If the automated prover fails (e.g., it exceeds the allocated time limit), we consider the evaluation to be unsuccessful.

## 4 EXPERIMENTS

### 4.1 DATASET AND EVALUATION

We evaluate our method on the miniF2F dataset (Zheng et al., 2022). The dataset contains the *formal statements* of 488 problems from high-school mathematical competitions, written in three formal languages: Lean, HOL-Light, and Isabelle. They are split into a valid set and a test set, composed

of 244 problems each. In this work, we choose to experiment with Isabelle for three reasons: (1) Isabelle's proof corpus is one of the largest among interactive theorem provers, conducive to the language models' mastery of its syntax; (2) Isabelle supports the declarative proof style (detailed discussion in Appendix A), enabling formal proof sketches (Wiedijk, 2003) which are central to our method; (3) although automated proving tools are available in other interactive theorem provers, none are as developed and effective as Sledgehammer (Paulson, 2010) in Isabelle for proving conjectures.

The miniF2F dataset is comprised of problems from three source categories: (1) 260 problems sampled from the MATH dataset (Hendrycks et al., 2021); (2) 160 problems from actual high-school mathematical competitions (AMC, AIME, and IMO); (3) 68 crafted problems at the same difficulty level as (2). We employ three methods to obtain informal statements and proofs from these sources. For source (1), we access the informal statements and proofs from the MATH dataset; for (2), we retrieve their informal statements and proofs from the AOPS website [1]; and for (3), we manually write down their informal statements and proofs. Thus we gather a parallel set of 488 informal statements, informal proofs, and formal statements. This dataset provides the informal statements and proofs for our experiment in the human-as-informal-proof-writer setting and will be released upon publication.

Our task is to generate formal proofs for problems as they are formally stated in miniF2F. We consider a proof valid if and only if it (a) does not contain "cheating" keywords (`sorry` and `oops`) that exit a proof without completing it, and (b) Isabelle is able to verify the corresponding formal statement with the proof. We use the Portal-to-ISAbelle API by Jiang et al. (2021) to interact with Isabelle.

### 4.2 BASELINES

**Sledgehammer** As a baseline, we attempt to prove the formal statement directly with Sledgehammer, a popular proof automation tool in Isabelle. We use the default Sledgehammer configuration in Isabelle2021, including a 120-second timeout and the five automated theorem provers (Z3, CVC4, SPASS, Vampire, E). Appendix B gives a more thorough introduction to Sledgehammer.

**Sledgehammer + heuristics** Occasionally, Sledgehammer may fail without trying simple yet effective tactics. As a second, stronger baseline, we create an automated prover that tries 11 common tactics (`auto`, `simp`, `blast`, `fastforce`, `force`, `eval`, `presburger`, `sos`, `arith`, `linarith`, `auto simp: field_simps`) for high-school level algebra and number theory problems. If every attempted tactic fails, or times out after 10 seconds, it falls back to Sledgehammer.

**Language models for proof search** Finally, we include baselines which are representative of state-of-the-art neural theorem proving in Isabelle, specifically Thor (Jiang et al., 2022) and Thor with expert iteration on autoformalized data (Wu et al., 2022). The methods GPT-f with expert iteration (Polu et al., 2022), and HyperTree Proof Search (HTPS) (Lample et al., 2022) can solve 36.6% and 41.0% of the problems on miniF2F-test. However, they rely on the Lean theorem prover instead of Isabelle, which greatly influences the performance due to the different tactics and automation, and are not directly comparable to our method.

### 4.3 EXPERIMENTAL SETUP

The experimental code is at `github.com/albertqjiang/draft_sketch_prove`.

**Drafting** When informal proofs are generated, we condition a large language model on informal statements to sample 100 informal proofs per problem. Specifically, we use the Codex code-davinci-002 model (Chen et al., 2021) through the OpenAI API, and the 8B, 62B, and 540B versions of the Minerva model from Lewkowycz et al. (2022). We use greedy decoding for Codex and nucleus sampling (Holtzman et al., 2019) with temperature $T = 0.6$ and top_p $= 0.95$ for Minerva models.

**Sketching** For sketching, we manually prepare 20 autoformalization examples of the format (*informal statement*, *informal proof*, *formal statement*, *formal sketch*), to form a pool of high-quality demonstrations. Of these 20 examples, 10 are of the *algebra* type and 10 are of the *number theory* type. All examples are from the validation set of the miniF2F dataset and can be found in the supplementary materials. The sketches contain in-line comments as in Figure 2. If the name of the problem gives away its type (*algebra* or *number theory*), we only use examples of the corresponding type. We also ensure that the sampled few-shot examples do not contain the problem being solved. The prompt is 3 uniformly randomly sampled example from the pool concatenated with the current problem's (*informal statement*, *informal proof*, *formal statement*). We use this prompt to query the

---

[1]`https://artofproblemsolving.com/community`

Table 1: **Proving success rates on the miniF2F dataset with Isabelle** In the table are the success rates of four baselines, the DSP method with human and language model informal proofs, as well as three ablation studies, on the validation and the test sets of miniF2F. The highest success rates on each set are highlighted in bold. The performance difference between ablation studies and DSP with human informal proofs are enclosed in brackets.

| Success rate | miniF2F-valid | miniF2F-test |
| --- | --- | --- |
| *Baselines* | | |
| Sledgehammer | 9.9% | 10.4% |
| Sledgehammer + heuristics | 18.0% | 20.9% |
| Thor (Jiang et al., 2022) | 28.3% | 29.9% |
| Thor + expert iteration (Wu et al., 2022) | 37.3% | 35.2% |
| *Draft, Sketch, and Prove* | | |
| Human informal proof | 42.6% | **39.3**% |
| Codex informal proof | 40.6% | 35.3% |
| 8B Minerva informal proof | 40.6% | 35.3% |
| 62B Minerva informal proof | **43.9**% | 37.7% |
| 540B Minerva informal proof | 42.6% | 38.9% |
| *Ablations (with human informal statements and proofs)* | | |
| – In-line comments | 37.7% (−4.9%) | 36.5% (−2.8%) |
| – Informal proofs | 38.9% (−3.7%) | 34.0% (−5.3%) |
| – Automated provers | 32.8% (−9.8%) | 30.3% (−9.0%) |

same Codex model to get the desired proof sketches. We use deterministic greedy decoding and a maximum of 2048 tokens in the generated sequence. For all the experiments, unless stated otherwise, we control the total number of queries made to Codex per problem to be 100. This means 100 queries per human informal solution and one query per language-model-generated solution.

**Proving** To prove the conjectures left open by the formal sketch, we use the Sledgehammer + heuristics automated prover described in Subsection 4.2. We execute the automated prover on every open conjecture in the sketch to synthesize a formal proof that can be verified by Isabelle.

## 4.4 RESULTS

In Table 1, we display the proportion of successful formal proofs found on the miniF2F dataset with the interactive theorem prover Isabelle. The results include the four baselines described in Subsection 4.2 and the *DSP* method with human-written proofs and model-generated proofs. From the table, we can see that the automated prover with 11 additional heuristic tactics significantly increases the performance of Sledgehammer, boosting its success rate from 9.9% to 18.0% on the validation set of miniF2F and from 10.4% to 20.9% on the test set. The two baselines using language models and proof search (Thor and Thor + expert iteration) achieve success rates of 29.9% and 35.2% on the test set of miniF2F, respectively.

With informal proofs written by humans, the *DSP* method achieves success rates of 42.6% and 39.3% on the validation and test sets of miniF2F. A total of 200 out of 488 problems can be proved in this way. The Codex model and the Minerva (8B) model give very similar results in solving problems on miniF2F: they both guide the automated prover to solve 40.6% and 35.3% of problems on the validation and the test sets respectively. This is corroborated by Lewkowycz et al. (2022)'s observation that these two models have comparable performances in solving mathematical problems.

When we switch to the Minerva (62B) model, the success rates rise up to 43.9% and 37.7% respectively. Compared to human-written informal proofs, its success rates are 1.3% higher on the validation set and 1.6% lower on the test set. In total, the Minerva (62B) model is able to solve 199 problems on miniF2F, one fewer than with human proofs. The *DSP* method is effective in guiding the automated prover under both settings that we study: using either human informal proofs or language-model-generated informal proofs. *DSP* almost doubles the prover's success rate and results in a new state-of-the-art performance on miniF2F with Isabelle. Moreover, the larger Minerva model is almost as helpful as a human in guiding the automated prover in solving problems.

## 5 ANALYSIS

### 5.1 ABLATION STUDIES

**Ablation of in-line comments** To facilitate the alignment between the informal proofs and the formal proof sketches, we copy relevant segments of the informal proofs as in-line comments in

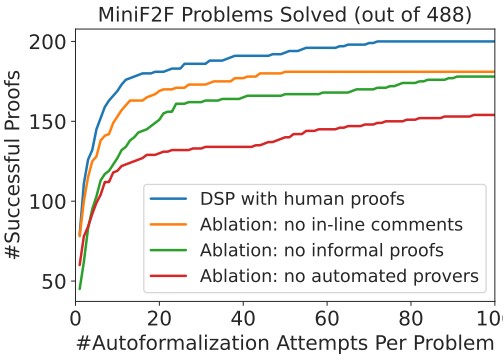 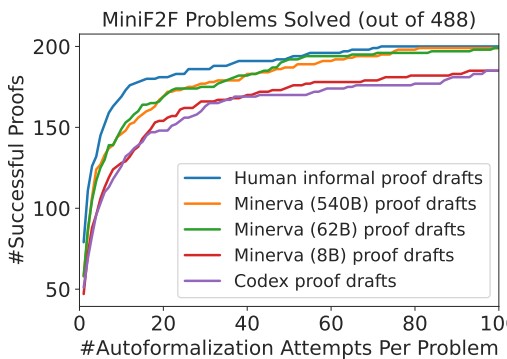

Figure 3: **Number of problems solved on miniF2F against the number of autoformalization attempts per problem. Left:** The figure displays the experiments carried out with the *DSP* method and three ablations on it. The curves represent the *DSP* method (blue), formal proof sketches without the in-line comments (orange), without informal proofs altogether (green), and without the automated provers (red). **Right:** The figure compares the experimental results with informal proof drafts written by humans (blue), the 540B Minerva model (orange), the 62B Minerva model (green), the 8B Minerva model (red), and the Codex model (purple).

the sketches. In the manually constructed prompt examples, these comments are prefixed to the corresponding Isabelle code blocks, as shown in Figure 2 (the text in red). We hypothesize that this technique is beneficial for large language models to synthesize formal sketches. To validate this hypothesis, we perform an ablation study by removing the in-line comments in the prompt examples before running the experiment. The results are displayed in Table 1. We find that without in-line comments, the success rates drop by 4.9% and 2.8% on the validation and test sets respectively. We conclude that having in-line comments is helpful for generating formal proof sketches.

**Ablation of informal proof drafts**   Drafting informal proofs is the first step of the *DSP* method. To investigate the necessity of this step, we perform an experiment of formal sketching and proving without informal proofs at all. Because formal proof sketches are written in the declarative proof style, they are fairly similar to the informal proof drafts already. Concretely, we remove the informal proofs and the in-line comments (because they are copied segments of the informal proofs) in the prompt examples. This removes the need for the informal proof writer, whether a human or a neural network. The results of this setup are shown in Table 1. It can be seen that the success rates on the validation and the test sets of miniF2F drop by 3.7% and 5.3% respectively compared to with human-written proofs. They are also inferior to success rates obtained with language-model-generated informal proofs. This demonstrates the importance of drafting informal proofs before sketching and proving.

**Ablation of automated provers**   Using an autoformalizer to generate proof sketches which are then completed by an automated prover is central to our method. The effect of utilizing an automated prover to close open conjectures in proof sketches is worth studying, so we conduct an ablation experiment for it. Namely, we replace the proof sketches in the prompt examples with complete formal proofs. The complete formal proofs still follow the declarative proof style, but do not contain any open conjectures. As a result, the large language model will also generate full proofs instead of sketches, and we directly check whether these generated proofs are valid. The results in this setup are presented in Table 1. The results reveal that without an automated prover to close open conjectures, the success rate on miniF2F decreases by 9.8% and 9.0% on the validation and test sets respectively. The drastic performance difference indicates the essential role of automated provers in our approach.

**Scaling properties of ablation studies**   To understand the effect of the ablations on the *DSP* method's scaling properties, we vary the number of autoformalization attempts per problem and plot the number of successful proofs found on the miniF2F dataset in Figure 3 (left). Three methods are contrasted: the original *DSP* method with human informal proofs, the *DSP* method without in-line comments, and the *DSP* method without sketching. It can be seen from the figure that with the original *DSP* method, the performance reaches a plateau (no new proofs are found) after 70 autoformalization attempts are made for each problem. For the ablation study with no in-line comments, the plateau is reached much faster, after around 50 autoformalization attempts. This method solves 181 problems in total. The ablation study without sketching can solve 154 problems on miniF2F. In comparison, with human informal proofs, only 7 autoformalization attempts are required to reach this performance.

---

**Statement:** Prove that the fraction $\frac{21n+4}{14n+3}$ is irreducible for every natural number $n$.

---

**Informal Proof (Minerva 62B):**
We must show that $\gcd(21n + 4, 14n + 3) = 1$.
The Euclidean algorithm gives

$$21n + 4 = 1 \cdot (14n + 3) + 7n + 1$$
$$14n + 3 = 2 \cdot (7n + 1) + 1.$$

Since $\gcd(7n + 1, 1) = 1$, we have $\gcd(21n + 4, 14n + 3) = 1$.

---

**Formal Proof:**

```
theorem imo_1959_p1:
  fixes n :: nat
  shows "gcd (21*n + 4) (14*n + 3) = 1"
proof -
  (* The Euclidean algorithm gives
  21n+4=1\cdot(14n+3)+7n+1
  14n+3=2\cdot(7n+1)+1. *)
  have c0: "21*n + 4 = 1*(14*n + 3) + 7*n + 1" <ATP> by auto </ATP>
  have c1: "14*n + 3 = 2*(7*n + 1) + 1" using c0 <ATP> by auto </ATP>
  (* Since \gcd(7n+1,1)=1, we have \gcd(21n+4,14n+3)=1. *)
  then have "gcd (7*n + 1) 1 = 1"
    using c1 <ATP> by auto </ATP>
  then have "gcd (21*n + 4) (14*n + 3) = 1"
    using c1 <ATP> by (smt (z3) BitM_plus_one ab_semigroup_add_class.add_ac(1)
    add.assoc c0 gcd.commute gcd_add2 gcd_add_mult mult_numeral_1
    numeral_One numeral_eq_Suc numerals(1) semiring_norm(3)) </ATP>
  then show ?thesis using c1 <ATP> by blast </ATP>
qed
```

Figure 4: **IMO proof guided by a Minerva informal proof** An informal proof of the International Math Olympiad problem `imo_1959_p1` generated by Minerva that leads to a successful formal proof. The steps enclosed by the *ATP* delimiters are generated by an automated prover and all other steps are generated by the DSP autoformalizer.

## 5.2 LANGUAGE-MODEL-GENERATED PROOFS

Our experiments demonstrated that model-generated informal proofs from Minerva and Codex can help guide a formal theorem prover. In this section, we analyze the properties of these proofs further. Since the Minerva (62B and 540B) models give the best overall performance on miniF2F, we focus on the informal proofs they produce in this section.

**Minerva helps solve one IMO problem** Interestingly, our approach manages to solve one problem from the International Mathematical Olympiad (`imo_1959_1`) with a Minerva-generated solution, but not with the human proof. For this problem, we present the successful Minerva-generated informal proof draft and the formal proof in Figure 4. We hypothesize that the reason behind this phenomenon is that human proofs might leave gaps between conjectures that are too difficult for automated provers to solve. On the other hand, the diversity in language model informal proofs makes some of them more amenable to automated provers. In Appendix C, we analyze the human and the Minerva informal proofs for this problem in greater detail. In Appendix D, we present a manual evaluation of Minerva proofs, and 3 more case studies comparing the human and Minerva informal proofs.

Is there a way to detect which Minerva proofs are correct, without human evaluation? For a preliminary investigation, we filter out all the problems that can be solved directly with the automated prover from the 50 and are left with 27 informal proofs. Of these 27, 21 are completely correct, 6 still contain small errors, but none are nonsensical. With this simple filter, we achieve a precision of $77.8\%$ and a recall of $72.4\%$ in identifying correct Minerva informal proofs.

**Scaling properties of human and Minerva proofs**    To understand the influence of different informal proof sources on the scaling properties of *DSP*, we plot the number of successful proofs found on miniF2F against the number of autoformalization attempts per problem in Figure 3 (right). Note that for each problem, we have 1 informal proof by a human and 100 informal proof drafts by each language model. The one human proof is used 100 times for formal proof sketch generation, while each language model proof draft is used only once. The 62B and the 540B models result in more successful proofs than the smaller (8B) Minerva model and the Codex model, consistently for any number of attempts. The 8B Minerva model and the Codex model behave similarly, both finding 185 proofs in the end. Informal proofs written by humans help solve more problems than those by Minerva models for $1 - 100$ autoformalization attempts. However, the difference is small (1 problem) when 100 are made.

Noticing that the number of successful proofs does not plateau for the Minerva-generated proofs, we investigate how further increasing the number of autoformalization attempts changes the number of problems solved for human-written and language-model-generated proofs. For each problem, we use 1 human informal proof and sample 200 sketches for it; we also use the same 100 informal proof drafts by the Minerva (540B) language model and sample 2 sketches for each draft. The total number of sketches per problem is 200 in both settings. We find that with human informal proofs, 203 theorems (106/97 on valid/test) have successful formal proofs, while with language-model-generated informal proofs, 209 (111/98 on valid/test) theorems have successful formal proofs after the same number of attempts. This suggests that the diversity in language-model-generated informal proofs can benefit the automated formalization process more than the "ground-truth" human proofs.

### 5.3    MEMORIZATION

This work utilizes two language models that have been trained on a large amount of internet data. Several prior works (Trinh & Le, 2018; Carlini et al., 2022) pointed out that such models can memorize some fraction of the data they encounter during training. For drafting informal proofs, we mainly experimented with Minerva. Lewkowycz et al. (2022, Section 5) discussed the memorization effects within Minerva and concluded that they could not find evidence that its abilities are due to memorization. For the autoformalization of proof sketches, the Codex (`code-davinci-002`) model was used. Its training data was collected before June 2021[2], at which time the miniF2F dataset had not been made public. So the model cannot benefit from memorizing the exact problems and proofs. Therefore, it is inappropriate to attribute the abilities of models used in this paper to memorization.

## 6    CONCLUSION

In this paper, we introduced *Draft, Sketch, and Prove (DSP)*, a novel approach that takes advantage of informal proofs to synthesize formal proofs. We demonstrated its feasibility and effectiveness by reaching state-of-the-art performance on the miniF2F dataset with the Isabelle theorem prover. Central to our method are formal proof sketches that mirror the high-level reasoning structures of informal proofs. Our ablations showed that the ability to automatically convert informal proofs to proof sketches is critical to the success of *DSP*.

Our *DSP* method differs fundamentally from previous applications of machine learning to formal proof synthesis in two aspects. Firstly, while most approaches in the field focus on improving proof search, our method seeks to construct the entire formal proof structure from the informal proof in one decoding operation. The task of the automated prover is then simplified to filling the gaps between intermediate conjectures. Secondly, while existing approaches operate exclusively on formal data, *DSP* by design benefits from informal proofs.

In this work, we utilized a purely symbolic automated prover to close the gaps in proof sketches. In the future, we aim to equip *DSP* with more powerful mechanisms, such as HyperTree Proof Search (Lample et al., 2022), to broaden the scope of provable theorems. Similar to AlphaCode (Li et al., 2022), we found that the number of generations is crucial for performance. The computational cost of the autoformalizer being a bottleneck in our method, we seek to develop approaches able to generate high-quality proof sketches more efficiently.

---

[2]https://beta.openai.com/docs/models/codex-series-private-beta

ACKNOWLEDGEMENTS

We thank Rui Yuan and Kunhao Zheng for helping with the informal solutions used in our dataset. We thank Christian Szegedy for his feedback on the early draft.

FUNDING DISCLOSURE

AQJ and WL are supported by the ERC Advanced Grant ALEXANDRIA (Project GA 742178).

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

# APPENDIX

## A    CONJECTURES AND THE DECLARATIVE PROOF STYLE

Interactive theorem provers such as Isabelle and Mizar use a *declarative* proof style (Syme, 1997), in which a proof is interleaved with conjectures and their corresponding proofs. Syme (1997) stated that the list of conjectures in a declarative proof should be analogous to a proof sketch found in a mathematical textbook and sufficiently convincing for the reader. In practice, ITP users often prove a theorem by writing down a list of conjectures (a "formal sketch"), then attempt to find a proof of each conjecture (fill a "gap") with an automated system.

## B    SLEDGEHAMMER

Sledgehammer (Paulson, 2010) is a powerful system that automates reasoning with the interactive theorem prover Isabelle. It works by flattening the goals encoded in the higher-order logic used by Isabelle/HOL into other logics (e.g., first-order logic) which can then be fed into automated theorem provers such as E [3], CVC4 [4], Z3 [5], Vampire [6], and SPASS [7]. If any of these automated theorem provers succeeds in finding the proof in their own corresponding format, Sledgehammer reconstructs the proof in Isabelle/HOL with certified provers (`metis`, `meson`, and `smt`), which is relatively more interpretable by humans.

As a practical example of using Sledgehammer, one can declare a conjecture in Isabelle/HOL: `have "4 dvd (a::nat) ⟹ 2 dvd a"` and call Sledgehammer immediately afterwards. If Sledgehammer succeeds, it will return a proof step that proves the conjecture. In this example, the step is `by (meson dvd_trans even_numeral)`, which uses the `meson` resolution prover and two facts: that the division relation is transitive and that 4 is an even number. If Sledgehammer does not find the proof or timeouts, it will report failure.

## C    A PROOF TO AN INTERNATIONAL MATHEMATICAL OLYMPIAD PROBLEM

With the Minerva-generated solutions, a proof to the problem `imo_1959_p1` is discovered. This is the first problem of the first ever International Mathematical Olympiad (IMO). The informal problem statement, Minerva-generated informal solution, and DSP's formal proof are shown in Figure 4.

In Figure 4, we can see that the autoformalizer in DSP (a large language model), copies over parts of the informal proof generated by Minerva as in-line comments to precede their corresponding formal proof blocks. The formal proof does not use the first sentence of the informal proof solution as it is already identical to the formal statement. We also notice that the large language model selects relevant premises after writing down the conjectures (the steps starting with `using`) despite not every premise is strictly needed.

The formal proof creates 5 conjectures (4 `have` statements and 1 `show` statement) which are all subsequently proved by our automated theorem prover. The step to prove the statement `have "gcd (21*n + 4) (14*n + 3) = 1"` involves 2 verified low-level provers `smt` and `z3` and 10 lemmas/facts from outside the scope of the language model. It is highly unlikely that either the large language model or the automated theorem prover can finish this proof on its own.

**Unsuccessful human-written proof.**    In contrast, the human-written informal proof of this IMO problem did not lead to a successful formal proof. The human-written proof is:

Denoting the greatest common divisor of $a, b$ as $(a, b)$, we use the Euclidean algorithm:

$$(21n + 4, 14n + 3) = (7n + 1, 14n + 3) = (7n + 1, 1) = 1$$

---

[3]https://wwwlehre.dhbw-stuttgart.de/ sschulz/E/E.html
[4]https://cvc4.github.io/index.html
[5]https://github.com/Z3Prover/z3
[6]https://vprover.github.io/
[7]https://www.spass-prover.org/download/index.html

It follows that $\frac{21n+4}{14n+3}$ is irreducible. Q.E.D.

A key difference between the Minerva proof and the human proof is the way that invoking the Euclidean algorithm is described. The Minerva proof explicitly writes out the results of the Euclidean algorithm (e.g. $21n + 4 = 1 \cdot (14n + 3) + 7n + 1$), which are translated into the sketch (*c1* in Figure 4). The human proof introduces new notation to express the results indirectly in terms of greatest common divisors, which ends up being less suitable for sketching. For example, below is a sketch generated with the human proof, which has a conjecture that is semantically incorrect and hence cannot be closed by the automated prover:

```
theorem
  fixes n :: nat
  shows "gcd (21*n + 4) (14*n + 3) = 1"
proof -
  have "(21*n + 4, 14*n + 3) = (7*n + 1, 14*n + 3)"
    ATP  (* <--- UNSUCCESSFUL *)
  also have "... = (7*n + 1, 1)"
    ATP
  finally show ?thesis
    ATP
qed
```

## D MORE ANALYSIS ON HUMAN AND MINERVA INFORMAL PROOFS

We analyze the relationship between the validity of the formal proofs and the correctness of the informal proofs. For our analysis, we randomly sample 50 Minerva proofs of different problems, which are then successfully converted to formal proofs. We then manually evaluate the correctness of these 50 informal proofs. Among them, 29 proofs (58%) are entirely correct, 16 are incorrect with a clearly identifiable incorrect step, and 5 "proofs" are nonsensical and simply rephrase the final conclusions of the problems.

Seeing that a total of $16 + 5 = 21$ incorrect informal proofs can lead to successful formal proofs, we study how they guide the automated formal prover despite having flaws themselves. The 21 proofs divide into 2 cases: In the first case, we find 13 problems for which the informal proofs are mostly ignored, and the automated prover can find proofs by itself; In the other 8 problems, although the informal proofs are wrong, the autoformalizer manages to *correct* them, either by ignoring the erroneous steps or by stating their correct versions in the formal proof sketches. This suggests that the autoformalizer has some understanding of the mathematical statements and is not merely translating them from an informal language to a formal language. It is robust to slight noises in its input.

We present 3 case studies comparing the human and Minerva informal proofs. Particularly, Figure 7 shows a completely correct example and one example of each pathological case.

**Statement:** Given that $f + 3z = 11$ and $3(f - 1) - 5z = -68$, show that $f = -10$ and $z = 7$.

- - - - - - - - - - - - - - - - - - - - - - - - - - - - - - - - - - - - - - - - - - - - - - - - - - - - - - - -

**Informal Proof (Minerva 62B):**
We can solve for $f$ in the first equation, obtaining $f = 11 - 3z$.
Then we can substitute this expression for $f$ into the second equation,
obtaining $3(11 - 3z - 1) - 5z = -68$.
Solving for $z$, we obtain $z = 7$.
Then, we can substitute this value of $z$ into the expression for $f$, obtaining $f = 11 - 3 \cdot 7 = -10$.

- - - - - - - - - - - - - - - - - - - - - - - - - - - - - - - - - - - - - - - - - - - - - - - - - - - - - - - -

**Formal Proof:**
```
theorem algebra_2varlineareq_fp3zeq11_3tfm1m5zeqn68_feqn10_zeq7:
  fixes f z::complex
  assumes h0: "f + 3*z = 11"
      and h1: "3*(f - 1) - 5*z = -68"
    shows "f = -10 \<and> z = 7"
proof -
  (* We can solve for f in the first equation, obtaining f = 11 - 3z. *)
  have c0: "f = 11 - 3*z"
    using h0
    by (auto simp: field_simps)   [ATP]
  (* Then we can substitute this expression for f into the second equation,
     obtaining 3(11 - 3z - 1) - 5z = -68. *)
  have c1: "3*(11 - 3*z - 1) - 5*z = -68"
    using h1 c0
    by auto   [ATP]
  (* Solving for z, we obtain z = 7. *)
  have c2: "z = 7"
    using c1
    by auto   [ATP]
  (* Then, we can substitute this value of z into the expression for f,
     obtaining f = 11 - 3 \cdot 7 = -10.*)
  have "f = 11 - 3*7"
    using c0 c2
    by auto   [ATP]
  then have c3: "f = -10"
    by auto   [ATP]
  show ?thesis
    using c2 c3
    by auto   [ATP]
qed
```

Figure 5: **Algebra example with Minerva informal proof.** An informal proof generated by Minerva that led to a successful formal proof. The autoformalizer generated a proof sketch containing all lines of the formal proof except for those with *[ATP]*. The sketch is structured according to the informal proof, containing five intermediate conjectures based on the informal proof. The autoformalizer generated in-line comments in the proof sketch (shown in red), which correctly identified an alignment between the formal and informal proofs.

**Statement:** For any integer $a$, show that $a^2 \equiv 0 \pmod 4$ or $a^2 \equiv 1 \pmod 4$.

---

**Informal Proof (Minerva 62B):**
Let $a = 2k$ for some integer $k$.
Then $a^2 = 4k^2$.
Since $4k^2$ is divisible by 4, we have $a^2 \equiv 0 \pmod 4$.
Now, let $a = 2k + 1$ for some integer $k$.
Then $a^2 = 4k^2 + 4k + 1 = 4(k^2 + k) + 1$.
Since $k^2 + k$ is an integer, $4(k^2 + k) + 1$ is not divisible by 4.
Thus, $a^2 \equiv 1 \pmod 4$.

---

**Formal Proof:**

```
theorem numbertheory_sqmod4in01d:
  fixes a :: int
  shows "(a^2 mod 4 = 0)
  \<or> (a^2 mod 4 = 1)"
proof (cases "even a")
case True
  (* Let a=2k for some integer k.
     Then a^2=4k^2.
     Since 4k^2 is divisible by 4,
     we have a^2 \equiv 0 \pmod{4}.*)
  then obtain k where "a=2*k"
    using evenE
    by auto   [ATP]
  then have "a^2 = 4*k^2"
    unfolding power2_eq_square
    by auto   [ATP]
  then have "a^2 mod 4 = 0"
    by auto   [ATP]
  then show ?thesis
    by auto   [ATP]
next
case False
  (* Now, let a=2k+1 for some integer k.
     Then a^2=4k^2+4k+1=4(k^2+k)+1.
     Since k^2+k is an integer,
     4(k^2+k)+1 is not divisible by 4.
     Thus, a^2 \equiv 1 \pmod{4}.*)
  then obtain k where "a=2*k+1"
   using oddE by auto   [ATP]
  then have "a^2 = 4*k^2+4*k+1"
   unfolding power2_eq_square
   by (auto simp: field_simps) [ATP]
  then have "a^2 mod 4 = 1"
   by presburger   [ATP]
  then show ?thesis
   by auto   [ATP]
qed
```

**Statement:** For any integer $a$, show that $a^2 \equiv 0 \pmod 4$ or $a^2 \equiv 1 \pmod 4$.

---

**Informal Proof (Minerva 62B):**
$a \pmod 4 \in \{0, 1, 2, 3\}$.
Using that for any natural number $k$, $a \equiv b \pmod 4$ implies $a^k \equiv b^k \pmod 4$, we have $a^2 \pmod 4 \in \{0, 1, 4, 9\}$.
Since $4 \equiv 0 \pmod 4$ and $9 \equiv 1 \pmod 4$, the result follows.

---

**Formal Proof:**

```
theorem numbertheory_sqmod4in01d:
  fixes a :: int
  shows "(a^2 mod 4 = 0)
  \<or> (a^2 mod 4 = 1)"
proof -
(* a \pmod 4 \in {0, 1, 2, 3}. *)
have c0: "a mod 4 \<in> {0, 1, 2, 3}"
  by auto   [ATP]
(* Using that for any natural number k,
   a \equiv b \pmod 4 implies
   a^k \equiv b^k \pmod 4,*)
have "a^2 mod 4 = (a mod 4)\<^sup>
      2 mod 4"
  by (smt (z3) numeral_eq_Suc power_mod)
  [ATP]
(* we have
   a^2 \pmod 4 \in {0, 1, 4, 9}. *)
also have "... \<in> {0, 1, 4, 9}"
  using c0
  by auto   [ATP]
(* Since 4 \equiv 0 \pmod 4 and
   9 \equiv 1 \pmod 4,
   the result follows. *)
finally show ?thesis
  by auto   [ATP]
qed
```

Figure 6: **Alternative proofs: Minerva (left) and human-written (right).** In both proofs, the formal sketch is structured based on the informal proof. The Minerva informal proof and its sketch break the proof into even and odd cases. These cases are not explicitly stated in the informal proof, and the formal sketch makes them explicit (*cases "even a"*). Each case has three conjectures that are directly based on the informal proof. The sketch excludes parts that are not needed in the formal proof, for instance "Since $k^2 + k$ is an integer, $4(k^2 + k) + 1$ is not divisible by 4". The human proof uses a different strategy than the Minerva proof, based on the facts that $a \pmod 4 \in \{0, 1, 2, 3\}$ and $a^2 \pmod 4 \in \{0, 1, 4, 9\}$. The sketch uses these as conjectures, synthesizes an intermediate step not in the informal proof, and excludes the last step of the informal proof.

**Statement:** For each plumbing repair job, Mr. Wrench charges $N$ dollars for coming out to the house plus $x$ dollars per hour that he works at the house. He charged \$97 for a one-hour repair job and \$265 for a five-hour repair job. What is his charge for a two-hour repair job? Show that it is \$139.

- - - - - - - - - - - - - - - - - - - - - - - - - - - -

**Informal Proof (Minerva 62B):**
Let $N$ be the number of dollars charged for coming out to the house and let $x$ be the number of dollars per hour that Mr. Wrench works. For the one-hour repair job, we have that the total cost is $N + x = 97$.
For the five-hour repair job, we have that the total cost is $N + 5x = 265$.
We subtract the first equation from the second equation to obtain $4x = 168 \Rightarrow x = 42$.
Plugging this back into the first equation, we obtain that $N = 55$.
Therefore, for a two-hour repair job, the total cost is $55 + 2x = 55 + 2(42) = \boxed{\$139}$.

- - - - - - - - - - - - - - - - - - - - - - - - - - - -

**Formal Proof:**

```
theorem mathd_algebra_160:
  fixes n x ::real
  assumes "n + x = 97"
  and "n + 5 * x = 265"
  shows "n + 2 * x = 139"
proof -
  (* We subtract the first equation
  from the second equation to obtain
  4x = 168 ⇒ x = 42. *)
  have "4 * x = 168"
  using assms by auto [ATP]
  then have "x = 42"
  by auto   [ATP]
  (* Plugging this back into
  the first equation, we obtain that
  N = 55. *)
  then have "n = 55"
  using assms by auto [ATP]
  (* Therefore, for a two-hour repair
  job, the total cost is
  55 + 2x = 55 + 2(42) = $139. *)
  then show ?thesis
  by (smt (z3) ⟨x = 42⟩)  [ATP]
qed
```

**Statement:** Show that for any real number $x$ and any natural number $n$, if $x > -1$, then $(1 + nx) \le (1 + x)^n$.

- - - - - - - - - - - - - - - - - - - - - - - - - - - -

**Informal Proof (Minerva 62B):**
This is true for $n = 1$. Now, suppose that this is true for $n$.
Then we have that

$$(1 + nx)^n = 1 + n^2 x + n^3 x^2 + \cdots + nx^n$$

$$\le 1 + nx + n^2 x^2 + \cdots + nx^n = (1 + x)^n$$

and

$$(1 + nx)^{n+1} = (1 + nx)^n (1 + nx)$$
$$\le (1 + x)^n (1 + x) = (1 + x)^{n+1}$$

Therefore, this is true for $n + 1$.

- - - - - - - - - - - - - - - - - - - - - - - - - - - -

**Formal Proof:**

```
theorem induction_1pxpownlt1pnx:
  fixes x :: real and n :: nat
  assumes "-1 < x"
  shows "(1 + n*x) \<le> (1 + x)^n"
proof (induct n)
case 0
then show ?case
by auto   [ATP]
next case (Suc n)
then show ?case
by (smt (z3) Bernoulli_inequality assms)
[ATP]
qed
```

**Statement:** Determine the remainder of 1529 (mod 6). Show that it is 5.

- - - - - - - - - - - - - - - - - - - - - - - - - - - -

**Informal Proof (Minerva 62B):**
By definition, $1529 \equiv 5 \pmod{6}$.

- - - - - - - - - - - - - - - - - - - - - - - - - - - -

**Formal Proof:**

```
theorem mathd_numbertheory_551 :
  "1529 mod 6 = (5::nat)"
by auto   [ATP]
```

Figure 7: **Three Types of Minerva proofs: correct proof (left), incorrect proof (right top), nonsensical proof (right bottom)** In the correct Minerva proof, the formal sketch is structured based on the informal proof and steps are well-aligned. In the incorrect Minerva proof, the step "This is true for $n = 1$" is corrected by Codex in the formal sketch to "case 0" which starts the base case with $n = 0$ since natural numbers include 0. This is an explicit correction made by Codex and makes a slightly incorrect Minerva proof formalized successfully. Lastly, the meaningless proof contains only a single statement without any calculation or justification. However, Codex also chooses to directly show the statement without any calculation. This suggests that the problem itself could be considered simple by Codex.

