# OpenReview forum: "Draft, Sketch, and Prove: Guiding Formal Theorem Provers with Informal Proofs"
_ICLR.cc/2023/Conference — ICLR 2023 notable top 5%_

### Official Review · Reviewer_AVNg · 2022-10-24

**Confidence:** 2
**Correctness:** 4
**Technical Novelty And Significance:** 3
**Empirical Novelty And Significance:** 3
**Recommendation:** 8

**Clarity, Quality, Novelty And Reproducibility:**

**Clarity**
- the paper was well written.

**Quality**
- this seems high quality, thorough work.

**Novelty**
- I have not seen this done before: to me, this seems a big advance.

**Reproducibility**
- no system details released.  Either code or a user interface would be helpful.

**Strength And Weaknesses:**

**Strengths**
See above

**Weaknesses**
1. I would like to see existing SOTA results.
1. if you could fit it, a plot of success v the number of steps in Appendix C would be nice.

**Next steps I would like to see in future work**
1. it would be particularly nice if DSP found new or 'better' proofs (akin to Newell, Shaw & Simon's Logic Theory Machine on Russell & Whitehead).
1. I would like to see consideration of multiple proofs of the same result: Nipkow's 2009 JAR article formalized two of Geanakoplos' three proofs of Arrow's impossibility theorem, to explore how easy different proofs mode could be formalized.  (I would _love_ to see style transfer eventually: prove this result in the style of von Neumann, or Erdős.)
1. Can the structure of proofs be better exploited?  For example, Hilbert-type proofs correspond to trees.  Isabelle-type proofs are more complex due to local assumptions.  (Was this the concluding reference to HyperTree?)
1. Thomas Hales' had a 'formal abstracts' project a few years ago, in which he sought to formalize abstracts of mathematical papers, easing searches across notation and terminology.  Could DSP help?

**Summary Of The Paper:**

The manuscript cleverly combines the ability of:
 1. theorem provers like Isabelle to accept `sorry` statements, in which unproved statements can be accepted as if true;
 2. large language models to fill in masked text.

It proposes a 'draft, sketch and prove' methodology that draws on the large corpus of informal (human) mathematics, converting it into a formal proof sketch with a large language model.  This sketch is then used to guide an automated theorem prover.

If I understand correctly, the methodology can use of erroneous (informal) proof sketches to generate correct proofs, as ML techniques generally make use of noisy training data to generate correct classifications.

The methodology seems to expand the class of informal mathematical problems that can be formalized and formally solved.

As the agenda seems ambitious and fruitful, I expect that it will lay the foundations for further advances, dropping the cost of mathematics, and extending the range of mathematical results for which we can establish proofs.

**Summary Of The Review:**

To me, this paper seems to represent in generating mathematical proofs at lower cost. I can see it launching significant research projects (q.v. possible future work, above).

---

> ### Author Response · Authors · 2022-11-13
> **Response to reviewer AVNg**
>
> We thank the reviewer for their helpful suggestions and encouraging feedback. Here we reply to specific points made by the reviewer:
>
> > I would like to see existing SOTA results.
>
> We have included the SOTA results on miniF2F achieved with the proof assistant Isabelle: Thor + expert iteration. We introduced it in Section 4.2 Baselines and compared with its performance in Table 1.
>
> > if you could fit it, a plot of success v the number of steps in Appendix C would be nice.
>
> Could you please clarify the contents for this plot? Appendix C contains a single proof to an IMO problem. We presume that by “success” you mean the number of successful proofs on miniF2F. What do you refer to by “the number of steps”?
>
> **For future work:**
> > it would be particularly nice if DSP found new or 'better' proofs (akin to Newell, Shaw & Simon's Logic Theory Machine on Russell & Whitehead).
>
> The style of the proof is dictated by the informal proof draft, since the sketching and proving part is “formalizing” without altering the essence of the proof. In our experience, we found that Minerva indeed comes up with different proofs to the same problem sometimes. To assess the quality/creativity of these proofs automatically would be an interesting avenue for new work.
>
> > I would like to see consideration of multiple proofs of the same result: Nipkow's 2009 JAR article formalized two of Geanakoplos' three proofs of Arrow's impossibility theorem, to explore how easy different proofs mode could be formalized. (I would love to see style transfer eventually: prove this result in the style of von Neumann, or Erdős.)
>
> We consider it a highly interesting pursuit to study how different proofs might have different compatibility with machine learning tools. It is not unlikely, with the abundance of data, that style transfer between proofs becomes a reality, given its success with visual art.
>
> > Can the structure of proofs be better exploited? For example, Hilbert-type proofs correspond to trees. Isabelle-type proofs are more complex due to local assumptions. (Was this the concluding reference to HyperTree?)
>
> We believe that the structure of proofs can indeed be better exploited. Currently language models mostly process sequences, discarding some of the richness in proof structure. HyperTree is more of a representation of the search structure, which is a data structure instituted by the search algorithm. We believe that they are different from proof structures, but there could be ways to reconcile them in the future.
>
> > Thomas Hales' had a 'formal abstracts' project a few years ago, in which he sought to formalize abstracts of mathematical papers, easing searches across notation and terminology. Could DSP help?
>
> We thank the reviewer for the pointer as this project was unknown to us. Indeed, making formalization of mathematical objects is one of the major goals of DSP. DSP can certainly help formalization given that appropriate domain data is gathered.
>
> > no system details released. Either code or a user interface would be helpful.
>
> In the supplementary material, we included the 20 examples used to form the few-shot prompt. Here we also give the pseudocode for sketching:
>
>     def sketch(examples: List[(str, str, str, str)], problem: (str, str, str)):
> 	    # examples: a list of length 20, containing tuples of (informal statement, informal proof, formal statement, formal sketch)
> 	    # problem: a tuple of (informal statement, informal proof, formal statement)
> 	    sampled_examples = random.sample(examples, n=3)
> 	    example_strings = [“ “.join(example) for example in sampled_examples]
> 	    input_prompt = “ “.join(example_strings) + “ “.join(problem)
> 	    return openai.Completion.create(
> 	        model=”code-davinci-002”,
> 	        prompt=input_prompt,
> 	        max_tokens=2048,
> 	        temperature=0
>         )
>
> With the supplied prompt examples, the pseudocode, and the informal dataset we will release upon the publication of the paper, it shall be straightforward to reproduce the experiments done with the informal proof drafts.
>
> We have made major updates to the paper, summarized in the general response. If you think our comment and update address your concerns, could you please consider raising your rating of the paper? Thank you!

---

> > ### Comment · Reviewer_AVNg · 2022-11-27
> > **good replies and a good paper: let's publish this!**
> >
> > > Could you please clarify the contents for this plot? Appendix C contains a single proof to an IMO problem. We presume that by “success” you mean the number of successful proofs on miniF2F. What do you refer to by “the number of steps”?
> >
> > The p.14 paragraph "The formal proof creates 5 conjectures ...  It is highly unlikely that ..." suggests, obviously, that the bigger the gaps left in the proof, the harder it is for DSP to fill them in.
> >
> > I'd wondered whether it would be insightful to plot that: how many (or which?) steps do you need to provide for DSP to succeed?

---

> > > ### Author Response · Authors · 2022-11-28
> > > **Additional plot**
> > >
> > > Dear reviewer AVNg,
> > >
> > > Thank you for your encouraging words.
> > >
> > > I have now fully understood the plot you are requesting. We have produced the plot of the histogram of the number of formal gaps in successful and failed sketches: https://ia801403.us.archive.org/3/items/test_20221213/test.pdf. The plot suggests that all successful sketches have $\leq 15$ open gaps, with the number of successes decreasing as the number of gaps increases. The failed sketches present a long-tailed distribution of formal gaps (failed sketches with $> 25$ gaps are grouped together for better visual effect). This is insightful as it suggests a trade-off between how detailed the high-level steps are and how many high-level steps there should be, to ensure the success of the formal proof.
> > >
> > > We thank the reviewer again for the suggestion of the plot as it provides valuable insights. We will add this in the final version of the paper.
> > >
> > > What else should we do for you to consider raising the rating of our paper?

---

> > > > ### Comment · Reviewer_AVNg · 2022-12-13
> > > > **unable to view plot**
> > > >
> > > > Thank you for producing that plot.
> > > >
> > > > I am unable to view it - I don't know whether the link expired, or whether my refusal to enable notifications from a site that I do not know was the problem.  (The original page advertised something called somewhat pornographic after spawning other pages.)
> > > >
> > > > Could you use a standard pastebin site?

---

> > > > > ### Author Response · Authors · 2022-12-13
> > > > > **Apologies and file re-uploaded**
> > > > >
> > > > > Please find the plot at https://ia801403.us.archive.org/3/items/test_20221213/test.pdf.
> > > > >
> > > > > I am deeply embarrassed and apologetic for the link shared earlier. I have edited the previous comment to alter the link also.

---

### Official Review · Reviewer_AYPt · 2022-10-24

**Confidence:** 5
**Correctness:** 3
**Technical Novelty And Significance:** 3
**Empirical Novelty And Significance:** 3
**Recommendation:** 8

**Clarity, Quality, Novelty And Reproducibility:**

The presentation is very clear and of good quality. The approach is a novel and elegant combination of existing approaches. I believe the results are reproducible if Minerva and Codex are both publicly available.

**Strength And Weaknesses:**

Strength:

+ The paper is well written and easy to follow. I enjoyed reading it.

+ The approach is technically sound, conceptually simple (which is good) and empirically working. Even better, the design of entire process is quite natural: first coming up with a proof in natural language and then turning it into a formal proof, organized in a way that subgoals can be easily automated, which is exactly what a human ITP expert would do.

+ The paper nicely assembles existing approaches from subareas of ML for theorem proving (i.e., theorem proving in natural language, autoformalization, and learning-based automation for ITP) to make a complete procedure that provides high-level automation for interactive theorem proving.

Weakness:

- The good performance is probably not that surprising. Both Minerva (used for drafting) and Codex (used for sketching) might have seen the problems somewhere on the internet, formal or informal version in different forms (e.g., comments or code from different provers, or math overflow etc). After all, the competetion problems are famous.

Minors:

- It is [reported](https://github.com/openai/miniF2F/issues/103) that some problems in MiniF2F are not well formalized and some are wrong. The performance should perhaps be interpreted carefully. But apparently this isn't the authors fault.

- In related work the authors wrote: "... perform search over the generated subgoals using powerful search methods such as MCTS". Recently there have also been approaches (e.g., [1][2]) proposed where learnable search could be useful for formal reasoning. Citations are missing.

[1] Wu et al. 2021, TacticZero: Learning to Prove Theorems from Scratch with Deep Reinforcement Learning

[2] Jonathan Laurent, André Platzer 2022, Learning to Find Proofs and Theorems by Learning to Refine Search Strategies: The Case of Loop Invariant Synthesis

**Summary Of The Paper:**

The paper proposed a novel and nice procedure for automating interactive theorem proving, which combines state-of-the-art approaches to ML for (interactive) theorem proving. The key idea is to have a proof written in natural language (either provided by a human user or generated by a language model) translated into a formal sketch which is a skeleton of a proof script that can potentially be verified by the underlying theorem prover when all the details are filled in. An automated theorem prover, which can either be a high-level tactic of the underlying system, or an external learning based system, is then used to close the gaps in the sketch.


**Summary Of The Review:**

The paper proposed a nice procedure for high-level automation for interactive theorem proving, integrating state-of-the-art approaches to ML for (interactive) theorem proving as its subroutines. The empirical evaluation validates the approach. I recommend accepting the paper.

---

> ### Author Response · Authors · 2022-11-13
> **Response to reviewer AYPt**
>
> We thank the reviewer for their helpful suggestions and encouraging feedback. Here we reply to specific points made by the reviewer:
>
> > The good performance is probably not that surprising. Both Minerva (used for drafting) and Codex (used for sketching) might have seen the problems somewhere on the internet, formal or informal version in different forms (e.g., comments or code from different provers, or math overflow etc). After all, the competition problems are famous.
>
> We agree with the reviewer that the memorization of large language models should be examined carefully. We need to point out the fact that the specific Codex model we used (code-davinci-002) was trained on data collected before the miniF2F problems were released. Before then, to the best of our knowledge, there has not been a collection of high-school competition problems formalized in Isabelle or Lean. Hence we do not think it is plausible that Codex has seen **formal** versions of the problems we deal with in the paper.
>
> For drafting, the original Minerva paper discussed the memorization issue in detail and found no evidence that its mathematical reasoning ability should be attributed to memorization. We updated the paper with a paragraph discussing the memorization problem in the new version.
>
> We emphasize that the main contribution of this paper is the Draft, Sketch, and Prove procedure, for which we can use different models for each part. The situation where we use human-drafted informal proofs is closer to the formalization practice, where usually an informal version of the proof is available. This situation also does not rely on Minerva at all.
>
> > It is reported that some problems in MiniF2F are not well formalized and some are wrong. The performance should perhaps be interpreted carefully. But apparently this isn't the authors’ fault.
>
> We noticed this problem too. We agree with the reviewer that this is not our fault. For a fair comparison with previous work, we used the same miniF2F benchmark. In fact, we manually examined the miniF2F dataset ourselves, and fixed the formal statements that are erroneous, according to the Github issue the reviewer points us to. During the process of fixing these problems, we found in the majority of cases, it is the conclusion that is formalized incorrectly and not the assumptions. This means that most of the incorrectly formalized problems are **not provable**. Consequently, this indicates that our success rate on the miniF2F benchmark is a conservative estimate of that when the errors are corrected.
>
> We will release the corrected dataset (with >50 fixes to formal statements) upon publication of this paper.
>
> > In related work the authors wrote: "... perform search over the generated subgoals using powerful search methods such as MCTS". Recently there have also been approaches (e.g., [1][2]) proposed where learnable search could be useful for formal reasoning. Citations are missing.
>
> We thank the reviewer for pointing out the missing citations. We have added the citations in the updated version.
>
> We have made major updates to the paper, summarized in the general response. If you think our comment and update address your concerns, could you please consider raising your rating of the paper? Thank you!

---

### Official Review · Reviewer_7zK6 · 2022-10-24

**Confidence:** 5
**Correctness:** 4
**Technical Novelty And Significance:** 4
**Empirical Novelty And Significance:** 4
**Recommendation:** 8

**Clarity, Quality, Novelty And Reproducibility:**

The paper is easy to follow. The idea is novel to convert informal statements, to informal proofs, to formal sketches, and finally to formal proofs.

**Strength And Weaknesses:**

This paper studies a promising direction of ATP,  to guide theorem provers using informal proofs.
The use of proof sketches is reasonable. The steps of prompting LLMs are technically sound.
The proposed approach requires no additional training on ground-truth proofs but achieves impressive results.

One question is that what if replace with the Sledgehammer + heuristics prover as the Thor prover? How much improvement can we get from Thor?

**Summary Of The Paper:**

This paper proposes to guide the formal theorem prover using informal proofs. The key idea is to convert the informal proofs to formal proof sketches by prompting LLM as a few-short learner. On the miniF2F-test dataset, this approach improves the performance of the Sledgehammer + heuristics prover of Isabelle from 20.9% to 39.3% with human informal proofs and 37.3 with LLM-generated informal proofs.

**Summary Of The Review:**

This paper proposes an interesting approach on how to make use of informal proofs for ATP. I recommend accepting this paper.

---

> ### Author Response · Authors · 2022-11-13
> **Response to reviewer 7zK6**
>
> We thank the reviewer for their helpful suggestions and encouraging feedback.
>
> > One question is that what if replace with the Sledgehammer + heuristics prover as the Thor prover? How much improvement can we get from Thor?
>
> We emphasize that our main contribution in this paper is the Draft, Sketch, and Prove procedure, which admits different mechanisms to carry out its stages. Hence we consider improvement in the automated provers an orthogonal line of work to ours. Having said that, the DSP method certainly admits Thor as the automated prover for the proving operation. Having Thor, or other methods that have better performances will improve the performance of the pipeline and we plan to integrate powerful provers in the pipeline as future extensions.
>
> We have made major updates to the paper, summarized in the general response. If you think our comment and update address your concerns, could you please consider raising your rating of the paper? Thank you!

---

### Official Review · Reviewer_9yAP · 2022-11-04

**Confidence:** 2
**Clarity, Quality, Novelty And Reproducibility:** 1. It is not completely clear how inf…
**Correctness:** 3
**Technical Novelty And Significance:** 3
**Empirical Novelty And Significance:** 3
**Recommendation:** 6

**Strength And Weaknesses:**

+ While using LLMs for auto-formalization seems novel enough, this is the first work where LLM is applied to generate the whole proof. The idea of guiding formal proof generation with informal proof sketches is novel. The experimental section justifies that sketching indeed helps generate more correct proofs. The study in this paper demonstrates that the model makes use of the alignment between formal and informal proofs. This is the main contribution of the paper.

The paper also contains some analysis of how the prover's performance depends on the correctness of informal proofs. It turns out that the prover can fix some simple errors in informal proof and even ignore the informal proof if it is incorrect. This observation is interesting and suggests that the model can perform some kind of formal mathematical reasoning.

The paper is generally well written, experimental section demonstrates state-of-the-art results on the MiniF2F dataset, ablation study justifies the usefulness of the guidance by the informal proofs and the alignment between informal and formal proofs. The approach is built on top of existing pre-trained models and seems easy to implement if the models are available.

Cons:
1. The examples play an important role and should be carefully constructed. In the paper, the authors try to use appropriate examples: first of all, all of them are from the MiniF2F dataset, secondly, whenever the problem type is known (from the problem name, e.g. it contains "number theory") they use the corresponding examples. So, the approach cannot be easily applied to other datasets.
2. More analysis of the model should be conducted. Figure 3, "Human informal proof drafts" implies that it is also important to have good examples as different proof attempts differ only in provided examples. How close are the examples of successful proofs to the statement compared to failed proofs?
It is also interesting to separate the effect of chosen examples from the effect of generated informal proofs.
For example, we can use the same 3 examples for all proof attempts and only change the informal proofs.
3. The method requires access to the LLM which is not widely available and limits the reproducibility of the approach.

**Summary Of The Paper:**

This paper proposes a large language models (LLM) to automatically generate formal proofs of mathematical statements. The idea is to use Codex or Minerva model to generate an informal proof (a proof in a natural language) and then use it as a sketch to generate a formal proof. Each claim in the sketch is a some kind of proposition, which is then proved by an off-the-shelf prover, like Sledgehammer. The approach is based on the ability of LLMs to do few-shot learning. A prompt to the model contains a formal statement to be proved, generated informal statement and informal proof, and several examples of tuples <informal statement, informal proof, formal statement, formal proof>. To provide appropriate examples, the authors gathered a dataset of parallel informal-formal proofs for the problems from the MiniF2F dataset and sample 3 random examples to make the prompt.

The approach was tested on validation and test splits of the MiniF2F dataset. In the experiments, the proposed approach outperforms the baselines. The ablation study demonstrates the benefit of using sketches and aligning them with formal proofs. Also, the results show that larger models have higher proof rates.

**Summary Of The Review:**

The paper introduces a novel idea to guide formal proof generation with generated informal proof using large language models in a few-shot setting. The idea was justified by the experiments and ablation study. However, more analysis of the proposed approach should be conducted to make a clear picture of what part of the approach and to what extent affects the final result.

---

> ### Author Response · Authors · 2022-11-13
> **Response to reviewer 9yAP (part 1)**
>
> We thank the reviewer for their helpful suggestions and constructive proposal for ablation study. We are currently conducting the experiment suggested and will update the reviewer next week when we have the results. Next we reply to some specific points raised by the reviewer.
>
> > The examples play an important role and should be carefully constructed. In the paper, the authors try to use appropriate examples: first of all, all of them are from the MiniF2F dataset, secondly, whenever the problem type is known (from the problem name, e.g. it contains "number theory") they use the corresponding examples. So, the approach cannot be easily applied to other datasets.
>
> Below we specify the procedure to apply the DSP approach to a formal mathematical reasoning dataset other than miniF2F and argue that the procedure is straightforward and it is feasible to apply it to other datasets.
> Examine the new dataset and manually prepare 20 examples of (informal statement, informal proof, formal statement, formal proof) tuples for problems from the dataset. The time it takes to prepare these depends on the complexity of the dataset. For comparison, writing the examples for miniF2F-level high-school competition problems took one author two days to complete. These examples are available as supplementary materials.
> Use the DSP approach with the new tuples as prompt examples.
> Usually formal mathematical reasoning corpora are organized by their domains(e.g., the Archive of Formal Proofs for Isabelle and mathlib for Lean), so picking representative problems that cover key concepts can be done conveniently and principally.
>
> > More analysis of the model should be conducted. Figure 3, "Human informal proof drafts" implies that it is also important to have good examples as different proof attempts differ only in provided examples. How close are the examples of successful proofs to the statement compared to failed proofs? It is also interesting to separate the effect of chosen examples from the effect of generated informal proofs. For example, we can use the same 3 examples for all proof attempts and only change the informal proofs.
>
> For each experiment conducted, we produce 100 autoformalization attempts for each of the 488 problems from miniF2F. So there are 48800 pairs of (prompt, problem) combinations and therefore a manual examination and analysis of the influence of different proof attempts is difficult to produce. The case study 3 of the autoformalization of statements paper [1] suggests that the autoformalization is more likely to succeed if the examples contain the definitions and concepts required. From our experience, this observation extends to the autoformalization of proofs: if the examples contain similar concepts and strategies required in the proof, it is more likely to succeed.
>
> We also thank the reviewer for suggesting the experiment with fixed instead of randomly sampled examples. The experiment can clearly demonstrate the importance of good examples quantitatively and is under way. We will make the update next week, when the results come out.
>
> We will update the paper to emphasize this point in the camera-ready version (due to page limit, we cannot make this addition in the rebuttal version).
>
> > The method requires access to the LLM which is not widely available and limits the reproducibility of the approach.
>
> It is unfortunate that the LLMs are not as accessible as we wish. However, to conduct the experiments with the human informal proof drafts in this paper, one only needs one LLM: OpenAI Codex. One can apply for access through the form OpenAI provides: https://share.hsforms.com/1GzaACuXwSsmLKPfmphF_1w4sk30?. In the authors’ experience, the application process is fairly fast (takes ~ half a week) and using Codex to do inference is completely free.

---

> > ### Author Response · Authors · 2022-11-13
> > **Response to reviewer 9yAP (part 2)**
> >
> > > It is not completely clear how informal proofs are generated. How input prompt is constructed? Does it include examples or does it consists of just the formal statement? Is it fixed across proof attempts or each attempt uses different informal proof? It is mentioned that greedy decoding is used for the Codex model. Does it allow generating different outputs for a fixed input prompt (usually greedy decoding provides deterministic output)?
> >
> > We thank the reviewer for pointing out that these points are not clear enough from the paper. We have emphasized them in the new version. Here we also provide an answer to each individual question:
> >
> > **How informal proofs are generated:** The human informal proofs come from three sources: (a) the MATH dataset (b) the AOPS website (c) we manually wrote them. The generation process is described in detail in the second paragraph on page 5. The Minerva-generated informal proofs are sampled from the Minerva language model without the majority voting procedure. For more details, we refer the reviewer to the Minerva paper as it is covered more thoroughly there: https://arxiv.org/abs/2206.14858.
> > **How input prompt is constructed:** We manually prepared the 20 input examples of the format specified (informal statement, informal proof, formal statement, formal sketch). We concatenate the 3 randomly sampled examples with the (informal statement, informal proof, formal statement) of the problem at hand as the input prompt. This is also covered in section 4.3 "sketching".
> > **Greedy decoding**: Greedy decoding, i.e., decoding the token with the maximum probability,  is used. This means that when the input prompt is fixed, the output is determined and will not change between attempts.
> >
> > > It is interesting to compare the computational resources required for one proof search. While the approach makes much fewer calls to the LLM generator, we have to run Sledgehammer several times during the proof search, so it is unclear whether the approach reduces the computation time.
> >
> > We need to emphasize that improvement in solving rate is the main contribution of this paper, and not reduction in computation resources. Nevertheless, we agree with the reviewer that comparison on computational resources is interesting. However, due to the differences in model sizes, hardware, and inference-time techniques, it is non-trivial to directly compare our approach with pure language-model-based proof search. Here we offer a comparison in wall-clock time of conducting one proof search with three approaches: (1) Sledgehammer (2) HTPS [2], a Monte Carlo Tree Search-like technique for proof search (3) DSP with human informal proofs.
> >
> > For optimal performance, Sledgehammer takes 2 mins to do one proof search. Sledgehammer only requires CPU cores and no hardware accelerator is needed.
> > From [2], we can calculate that for HTPS, 610 searches can be conducted in one day on 1 GPU on average, which is ~25 searches/hour. 20 searches are done in parallel, so one single proof search takes ~50 minutes. A hardware accelerator (GPU) is required throughout the search. The authors of [2] confirmed that this estimate is plausible.
> > For DSP with human informal proofs, to conduct one proof search, we make one call to OpenAI Codex, which takes a maximum of 20s to complete. A maximum of 20 Sledgehammer calls are then made, each with a 30s time limit. So the total wall-clock time of one proof search with DSP is 20s + 30s * 20 = 620s ~= 10 mins. Notice that only the first 20s require a hardware accelerator (through the OpenAI API).
> >
> > To summarize, for one proof search, DSP is slower than Sledgehammer but ~5 times faster than HTPS, with much less hardware accelerator time.

---

> > > ### Author Response · Authors · 2022-11-13
> > > **Response to reviewer 9yAP (part 3)**
> > >
> > > > Another interesting question here is data leakage. From the paper (Wu et al. 2022. Autoformalization with large language models) we can conclude that it is unlikely that the MiniF2F dataset was in the training dataset of Codex, but there is still a possibility that informal statements and proofs are contained in the training dataset of Codex.
> > >
> > > We agree with the reviewer that informal statements and proofs are contained in the training dataset of Codex. However, we do not think this affects the performance validity of our approach, for the following reason: we mainly use Codex in the sketching operation, with its **input** being informal statements and proofs + formal statements and its **output** being the formal sketches. That is to say, although the input might have data leakage, **the output is not leaked**. In fact, the formal sketches of miniF2F problems do not exist prior to this paper. Hence we do not think data leakage poses a problem.
> > >
> > > We have also updated the paper with a paragraph discussing memorization.
> > >
> > > We have made major updates to the paper, summarized in the general response. If you think our comment and update address your concerns, could you please consider raising your rating of the paper? Thank you!
> > >
> > > [1] Wu, Y., Jiang, A. Q., Li, W., Rabe, M. N., Staats, C., Jamnik, M., & Szegedy, C. (2022). Autoformalization with large language models. arXiv preprint arXiv:2205.12615.
> > >
> > > [2] Lample, G., Lachaux, M. A., Lavril, T., Martinet, X., Hayat, A., Ebner, G., ... & Lacroix, T. (2022). HyperTree Proof Search for Neural Theorem Proving. arXiv preprint arXiv:2205.11491.

---

> ### Author Response · Authors · 2022-11-19
> **Friendly reminder**
>
> We have posted a new official comment, detailing the result and the analysis of an ablation study you suggested. We want to thank you again for suggesting it and hope to hear your response to our rebuttal soon.

---

### Author Response · Authors · 2022-11-13
**General response to reviewers and paper update summary**

We want to thank all reviewers for their time and energy. Their feedback has been hugely helpful and we truly appreciate it.

We have updated the paper since the paper was submitted. We summarize the major changes here. In the paper, the main updates are also highlighted in blue.
1. We added the results obtained with the 540B Minerva model, boosting the test success rate on miniF2F with language model proof drafts from 37.7% to 38.9%. We also experimented with 2 formal proof sketch attempts per draft with Minerva 540B which yields 209 proofs in total.
2. We performed an additional ablation study on the informal proof drafts.
3. We added a discussion paragraph addressing the memorization issue.
4. Various improvements on writing.

---

### Author Response · Authors · 2022-11-19
**Additional ablation study results**

Per the suggestion of reviewer 9yAP, we conducted an additional ablation study. Here is the summary of the study:

- Setting: Instead of randomly sampling 3 examples from a pool of 20, we use a fixed set of 3 examples for prompting all sketching operations. We randomly took 3 examples from the same pool and performed 1 sketching operation for each problem with them, since the fixed set of examples + greedy decoding guarantees the same generated output every time.
- Hypothesis: The 20 examples provide diversity for the sampled output. The 3 randomly selected examples are of similar quality and should produce a similar number of successful proofs to the main experiment, but with 1 autoformalization attempt.
- Results: For the 488 miniF2F problems, we found 96 successful proofs.
- Analysis: With DSP and human informal proofs, 1 autoformalization attempt per problem results in 77 formal proofs. The result obtained in this ablation study is indeed comparable to the main method with 1 autoformalization attempt. However, the deterministic output limits its further scaling with respect to the autoformalization attempts.
- Conclusion: The results obtained with the DSP method do not rely on **particularly good** examples in the prompt, therefore the method is not brittle in that aspect. The 20 examples in the pool, however, do enable us to explore the advantage of diversity in prompts.

---

### Decision · Program_Chairs · 2023-01-20

**Decision:**

Accept: notable-top-5%

**Justification For Why Not Higher Score:**

I have recommended the highest score.

**Justification For Why Not Lower Score:**

This is an important contribution to automated reasoning using large language models.  It addresses the issue of logic which is largely believed to be a (or the) central problem with LLMs.

**Metareview: Summary, Strengths And Weaknesses:**

This paper describes an approach to both auto-formalization (translating informal mathematics to formal logical notation) and to proving the "proof sketches" that result from auto-formalizing informal proofs.  The empirical results are significant.  The reviewers most familiar with the area are gave strong reviews.  The paper directly addresses the core problem of logic and reasoning which is a fundamental problem with large language models.

A weakness is the use of Codex which has (slightly) restricted access.

**Note From Pc:**

if the above contains the word "oral" or "spotlight" please see: "oral" presentation means -> notable-top-5% and "spotlight" means -> notable-top-25%. As stated in our emails, we are disassociating presentation type from AC recommendations